# Surgical Stabilisation of Traumatic Rib Fractures with Chronic, Residual Type A Aortic Dissection

**DOI:** 10.3390/healthcare9040392

**Published:** 2021-04-01

**Authors:** Kieran J. Matic, Rajkumar Cheluvappa, Selwyn Selvendran

**Affiliations:** 1Department of Surgery, St George Hospital, Kogarah, NSW 2217, Australia; kieranmatic@gmail.com (K.J.M.); tselvendran@hotmail.com (S.S.); 2Australian Catholic University, Watson, ACT 2602, Australia

**Keywords:** aortic dissection, rib fracture, rib stabilisation, stabilisation of rib fractures

## Abstract

Surgical stabilisation of rib fractures (SSRF) reduces morbidity and mortality. However, its impact in complicated cases, particularly those with underlying thoracic pathologies, is of continued interest. Electronic records were retrospectively reviewed after obtaining informed consent from the patient. This case report details a patient with chronic, residual, Stanford Type A aortic dissection (AD) who had multiple left-sided rib fractures with a flail segment after being struck by a bicycle. The preoperative computed tomography (CT) of the patient’s chest showed that the sixth posterior rib fracture location was just ~13 mm from the false lumen of the aorta. As the patient had poor respiratory output and persistent pain, SSRF was not performed on the posterior sections. However, the anterior third to seventh rib fractures were plated. The patient recovered fully, with reduced pain and improved respiratory function. This is the first report describing the benefits of SSRF with AD or major thoracic pathologies. Further research into the benefits of SSRF in specific thoracic pathologies may lead to improved patient outcomes. This may require the creation of profiles of patient cohorts with relevant clinical history to determine if SSRF may benefit patients with specific thoracic pathologies.


**Core Tip:**


○Surgical stabilisation of rib fractures (SSRF) reduces morbidity and mortality;○The effects of SSRF in patients with underlying chronic aortic dissection are unknown;○In a patient with chronic, residual, Stanford type A aortic dissection; SSRF improved respiratory function and reduced pain;○This is the first case report describing the effects of SSRF in chronic or residual type B aortic dissection.

## 1. Introduction and Literature Review

Aortic dissection (AD) is a life-threatening condition caused by intramural haemorrhage from a tear in the tunica intima causing a false lumen, which may lead to aortic rupture, cardiac tamponade, and/or end organ failure [1]. Although AD is rare, with an incidence of 4 per 100,000 in New South Wales (Australia), the 30-day mortality rate is 35% [2].

AD is usually classified using 3 clinical features/criteria [3]:(1)the site of dissection (the Stanford criteria), which is classified as either type A (ascending aorta or aortic arch) or type B (descending aorta);(2)the direction of the dissection, which can be defined as either retrograde (towards the aortic root) or anterograde (towards the iliac vessels); and(3)the duration of AD, which can be classified as either acute (<2 weeks) or chronic (>2 weeks).

Traumatic rib fractures leading to aortic injury or threatened aortic injury are rare [4]. Previous case reports have reported rib fractures causing aortic laceration, aortic graft laceration, or threatened laceration [4,5,6,7,8,9,10]. A case of aortic laceration in a patient undergoing surgical stabilisation of rib fractures (SSRF) has been reported [11]. There are limited reports on SSRF in patients with underlying AD. A recent case series included a patient with a traumatic Stanford Type B AD who had undergone a SSRF [12]. However, the case presented in this paper is the first report of a patient undergoing SSRF with a fracture in proximity to a known chronic Stanford type A AD.

## 2. Methods

Informed consent was obtained from the patient at St George Hospital, Sydney, NSW, Australia. Pertinent electronic medical records were retrospectively reviewed including clinical documentation, radiological imaging, and pathology results.

## 3. Case Details

A 66-year-old man presented to the emergency department after being struck by a cyclist. He complained of left-sided chest, shoulder/hip pain, and post-traumatic amnesia. He had a past medical history of hypertension, hypercholesterolaemia, and chronic, residual, Stanford Type A AD, which was repaired in 2014. On examination, he was haemodynamically stable with a brachial blood pressure of 115/71 mmHg, no significant blood pressure differences between arms, a respiratory rate of 22 breaths per minute, and a pulse oximeter saturation of 95% on 2 litres nasal prong oxygen. He had a tender left chest wall posteriorly and laterally, with extensive subcutaneous emphysema over his chest wall and neck bilaterally. He had reduced air entry in his left chest. His had dual heart sounds without murmurs. A computed tomography (CT) scan of his chest showed a comminuted left clavicle fracture, left rib fractures from the second to the ninth, with flail segments from the third to the ninth, and a moderate left pneumothorax (Figure 1 and Figure 2). The sixth posterior rib fracture was only 13 mm away from the aorta (Figure 1 and Figure 3A). The CT also demonstrated that the residual AD had increased in size since the original diagnosis in 2014. A CT of his brain and facial bones showed no new intracranial pathologies or fractures.

A 28-gauge French intercostal catheter was inserted into the patient’s left fifth intercostal space and connected to an underwater seal. The water in the water seal displayed oscillations. The patient was started on high flow nasal prongs at 30 L/min with 30% oxygen and intravenous morphine patient-controlled analgesia. On day 1 of his admission, he was able to perform an incentive spirometry of approximately only 250 mL. After discussion of the patient’s condition and clinical history with a cardiothoracic surgeon, it was decided that SSRF would not be performed in the posterior sections. Owing to the patient’s poor respiratory output and persistent pain, the decision was made to perform SSRF on the anterior rib fractures.

The patient underwent surgical stabilisation of his fractures on day 6 post-admission. He had a general anaesthetic with double lumen endotracheal tube for single lung ventilation. The patient was positioned in the left lateral position with a left flexed arm. A horizontal subpectoral incision was performed over the sixth rib. The anterior fractures were identified and lung adhesions to fractured ribs were mobilised gently.

Intraoperative thoracoscopy was performed with extreme care. Whilst using intraoperative thoracoscopy to identify the relative positions of the posterior rib fractures and the AD site, it was crucial to carry out the procedure with precision and meticulous care owing to the risk of AD rupture. The anterior third to seventh ribs were reduced, plated, and screwed. A paravertebral block was inserted under direct vision.

There were several technical challenges related to this SSRF process. Intensive general anaesthesia monitoring was necessary because of the complex issues involved. The patient’s blood pressure needed to be constantly monitored, steadied, and maintained because of the major risk of AD rupture. The monitoring needed to be done both intraoperatively and postoperatively. Whilst manipulating the rib fractures of the flail segment anterolaterally, meticulous care was needed to avoid further posterior rib or rib fragment displacement towards the AD site.

After fixation, two 19 French intercostal catheters were inserted before closing. The thoracic wall muscles and subcutaneous fat were approximated with 2-0 Vicyrl and the skin was closed with subcuticular 3-0 Monocryl. The patient was monitored in the high dependence unit for 2 days with no complications. On day 7 of admission, the left clavicle was fixed with surgical plating. Functional lung expansion improved substantially (>50%) on day 8 of his admission, which corresponded to day 2 after SSRF. On day 10 of his admission, corresponding to day 4 after SSRF, he was able to perform incentive spirometry adequately (1200 mL). He was discharged on day 14 of his admission with no opioid analgesia and without supplemental oxygen. He was followed up in an outpatient clinic after 4 weeks, where his pain was demonstrably well controlled (0/10 at rest and 3/10 on exertion), and his chest X ray and chest CT revealed good expansion of his chests with plates and screws in situ (Figure 3B).

## 4. Discussion

Traumatic rib fractures causing aortic injury or threatened aortic injury are rare [4]. It is rarer for traumatic rib fractures to have underlying aortic pathology as a pre-existing condition. Unfortunately, this surgical critical care topic (AD and SSRF) is a very circumscribed/specific field without much published data, information, or guidelines concerning management. However, there has been a resurgence in the use of SSRF in managing rib fractures. More documentation and research is required to determine specific patient profiles that may benefit from surgery [13]. Intraoperative thoracoscopy should be performed with extreme care to minimise the risk of AD rupture. Optimal intraoperative and postoperative blood pressure need to be meticulously maintained owing to the risk of AD rupture with blood pressure fluctuations. Our case study demonstrated reduced pain and increased lung function in a patient with chronic, residual AD who underwent SSRF.

## 5. Conclusions

Our case report demonstrates that SSRF could be performed safely on AD patients with multiple rib fractures and flail segment(s). We also show that SSRF may reduce pain and increase lung function in AD patients. Further research is required to determine which patients with other pathologies could benefit from SSRF.

## Figures and Tables

**Figure 1 healthcare-09-00392-f001:**
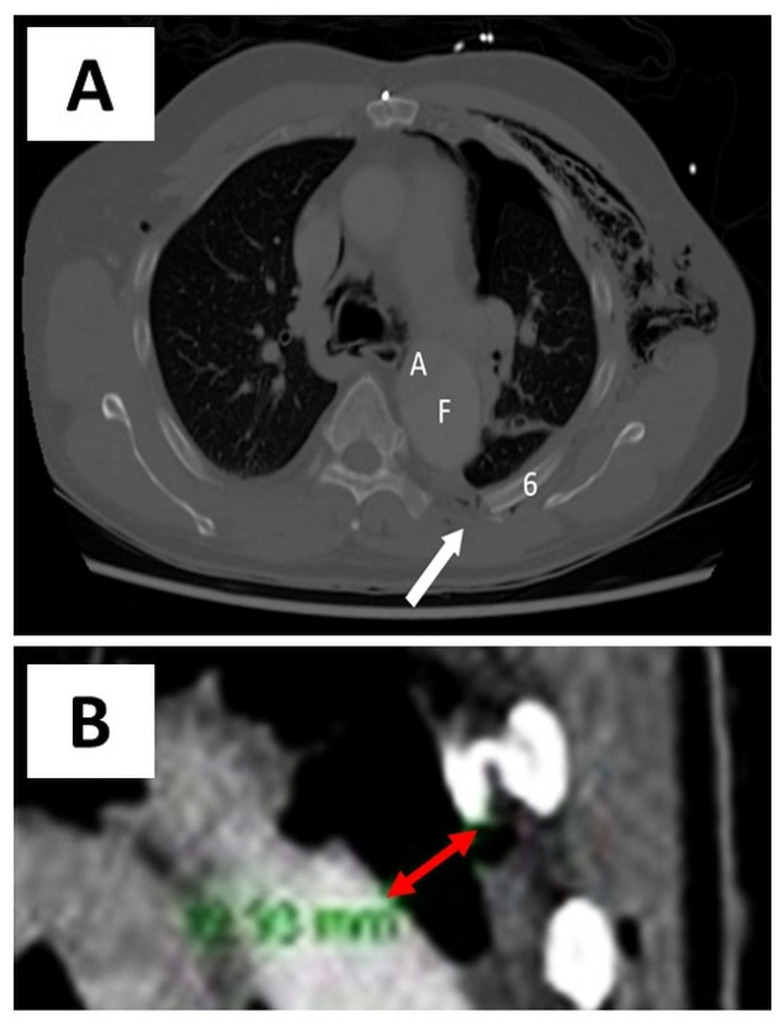
**Preoperative chest CT.** (**A**). Axial view—preoperative chest CT demonstrating the proximity of the fracture site to the aorta. Subcutaneous emphysema is also seen. (**B**). Section of longitudinal view—preoperative chest CT demonstrating the proximity of the fracture site to the false lumen of the aorta (~13 mm as indicated by the double-tipped red arrow). Arrow—fracture site; A—true lumen of aorta; F—false lumen of aorta; 6—sixth rib.

**Figure 2 healthcare-09-00392-f002:**
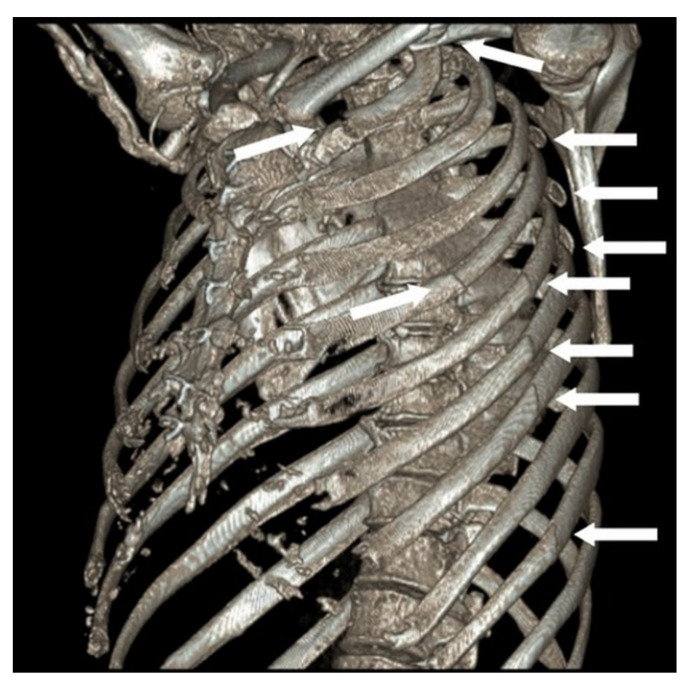
**Preoperative 3-dimensional CT reconstruction of chest.** Preoperative 3-dimensional CT reconstruction of chest—white arrows point to fracture sites.

**Figure 3 healthcare-09-00392-f003:**
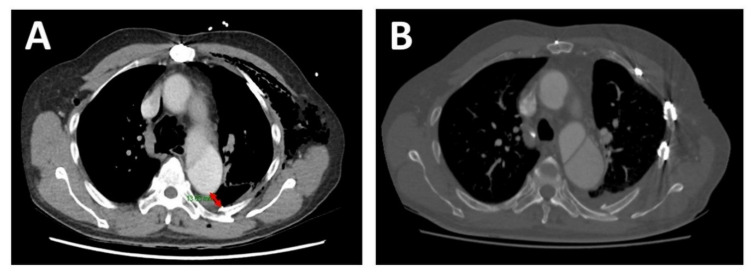
**Chest CT (axial view)—preoperative and 6-week postoperative images.** (**A**). Preoperative chest CT demonstrating the proximity of the sixth rib fracture site to the false lumen of the aorta (~13 mm as marked by the double-tipped red arrow). (**B**). 6-week postoperative chest CT showing in situ RibFix^TM^ plates.

## Data Availability

Data supporting the reported results can be found in the hospital records.

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
