# Peer review of "Surgical Stabilisation of Traumatic Rib Fractures with Chronic, Residual Type A Aortic Dissection"

_healthcare, 2021, doi:10.3390/healthcare9040392_

Round 1

Reviewer 1 Report

The manuscript titled “Surgical stabilization of traumatic rib fractures with chronic type A aortic dissection” by  K J Matic and co-workers, reports on a case report of  a patient with a residual chronic Stanford Type A aortic dissection (AD) who had multiple left sided rib fractures with a flail segment after being struck by a bicycle. An anterior surgical stabilization of rib fractures was performed on day 6 post admission. There were no postoperative complications and the patient was discharged on day 14 of his admission.  

Congratulations to the authors for the proposed work. The results of this case report are interesting, and the study is well done. This case demonstrated that surgical treatment of ribs fractures can be performed safely also on patients with aortic or other thoracic diseases.

In our opinion the authors should explain better in the title and in the text that the patient had a residual type A aortic dissection.

Author Response

The manuscript titled “Surgical stabilization of traumatic rib fractures with chronic type A aortic dissection” by  K J Matic and co-workers, reports on a case report of  a patient with a residual chronic Stanford Type A aortic dissection (AD) who had multiple left sided rib fractures with a flail segment after being struck by a bicycle. An anterior surgical stabilization of rib fractures was performed on day 6 post admission. There were no postoperative complications and the patient was discharged on day 14 of his admission.  Congratulations to the authors for the proposed work. The results of this case report are interesting, and the study is well done. This case demonstrated that surgical treatment of ribs fractures can be performed safely also on patients with aortic or other thoracic diseases.

  • Thank you very much.

In our opinion the authors should explain better in the title and in the text that the patient had a residual type A aortic dissection.

  • Thank you for your valuable feedback. Towards this, we have added the word “residual” to the title of the amended manuscript
  • We have also added the words “chronic, residual” or “residual” to the abstract, the core tip, and throughout the text of the amended manuscript.

Reviewer 2 Report

This is an interesting case of rib fixation in a patient with residual aortic dissection after a previous repair of the ascending aorta. The case is well described and the images are good. I only have minor comments.

1: Could you please better describe the technical challenging related to the rib fixation procedure and specifically to the presence of a concomitant residual AD?

2: Since the conclusion is that this procedure can be performed safely in the presence of AD, I would encourage the authors to provide suggestions on indication and procedural steps in the context of AD, especially when rib fragments are so close to the aorta. This might be helpful in defining how to perform safely this procedure in these patients.

3: The authors provide a pain score 4 weeks after surgery: it would be nice to understand how quickly the patient regain a fully functional lung expansion. Can the authors provide more information on the immediate postoperative recovery, in terms of functional status?

4: in row 108 I read: "his chest X-ray revealed good expansion of his chests with plates and screws in situ (Figure 3B)", but figure 3B shows a CT scan. Please amend the text accordingly.

5: the discussion should be expanded.

Author Response

This is an interesting case of rib fixation in a patient with residual aortic dissection after a previous repair of the ascending aorta. The case is well described and the images are good. I only have minor comments.

  • Thank you very much

1: Could you please better describe the technical challenging related to the rib fixation procedure and specifically to the presence of a concomitant residual AD?

2: Since the conclusion is that this procedure can be performed safely in the presence of AD, I would encourage the authors to provide suggestions on indication and procedural steps in the context of AD, especially when rib fragments are so close to the aorta. This might be helpful in defining how to perform safely this procedure in these patients.

Towards these, we have added the following two segments to the “Case details” section of the manuscript:

  • Intra-operative thoracoscopy was performed with extreme care. Whilst using intra-operative thoracoscopy to identify the relative positions of the posterior rib fractures and the AD site, it was crucial to carry out the procedure with precision and meticulous care owing to the risk of AD rupture.
  • There were several technical challenges related to this SSRF process. Intensive general anaesthesia monitoring was necessary because of the complex issues involved. The patient’s blood pressure needed to be constantly monitored, steadied, and maintained because of the major risk of AD rupture. The monitoring needed to be done both intraoperatively and postoperatively. Whilst manipulating the rib fractures of the flail segment anterolaterally, utmost care was needed to avoid further posterior rib or rib fragment displacement towards the AD site.

3: The authors provide a pain score 4 weeks after surgery: it would be nice to understand how quickly the patient regain a fully functional lung expansion. Can the authors provide more information on the immediate postoperative recovery, in terms of functional status?

  • Towards this, the sentence, “Functional lung expansion improved substantially (> 50%) on day 8 of his admission, which corresponds to day 2 after SSRF” has been added to the amended text.
  • The words “corresponding to day 4 after SSRF,” has been added to the sentence, “On day 10 of his admission, he was able to perform incentive spirometry adequately (1200 ml)” in the amended text.

4: in row 108 I read: "his chest X-ray revealed good expansion of his chests with plates and screws in situ (Figure 3B)", but figure 3B shows a CT scan. Please amend the text accordingly.

  • The text has been amended accordingly.

5: the discussion should be expanded.

  • The discussion has been expanded.